# Association between Proteinuria Status and Risk of Hypertension: A Nationwide Population-Based Cohort Study

**DOI:** 10.3390/jpm13091414

**Published:** 2023-09-20

**Authors:** Hyungwoo Lee, Moo-Suk Park, Min Kyoung Kang, Tae-Jin Song

**Affiliations:** Department of Neurology, Seoul Hospital, Ewha Womans University College of Medicine, Seoul 07804, Republic of Korea

**Keywords:** proteinuria, hypertension, prevention

## Abstract

Proteinuria is associated with cardiovascular disease. However, the relationship between changes in proteinuria status and hypertension remains unclear. This study aimed to explore the association between changes in proteinuria status and the risk of developing hypertension with the data from the Korean National Health Insurance Database. We included participants without prior hypertension history who underwent their first health examination in 2003–2004 and a second examination in 2005–2006. Based on their proteinuria status during these two examinations, participants were classified into four groups: the proteinuria-free, proteinuria-resolved, proteinuria-developed, and chronic proteinuria groups. The study outcome was the incidence of hypertension. The study included 935,723 participants followed for a median of 14.2 years (mean age: 40.96 ± 11.01, 62.5% male participants). During this period, 346,686 (37.1%) cases of hypertension were reported. The chronic proteinuria group had the highest hypertension risk, followed by the proteinuria-developed, proteinuria-resolved, and proteinuria-free groups (*p* < 0.001). Those who recovered from proteinuria had a lower risk of developing hypertension than those with chronic proteinuria (hazard ratio: 0.58; 95% confidence interval: 0.53–0.63, *p* < 0.001). In contrast, individuals who developed proteinuria had a higher risk of hypertension than proteinuria-free individuals (hazard ratio: 1.31; 95% confidence interval: 1.26–1.35, *p* < 0.001). Our findings suggest a significant association between proteinuria status changes and hypertension. Effective management of proteinuria may potentially decrease the risk of developing hypertension.

## 1. Introduction

The prevalence of hypertension is projected to reach 1.4 billion people by 2025 [1]. Given its significant association with various health problems including cardiovascular diseases, stroke, and mortality, it is crucial to identify preventive measures [2,3]. Many environmental factors, such as lifestyle choices and genetic predisposition, contribute to the occurrence of hypertension. Managing known risk factors, such as reducing salt intake, increasing dietary potassium, achieving a healthy weight, maintaining good oral health, and increasing physical activity, can help prevent hypertension [4,5]. However, further research is needed to identify other contributing factors and to develop more comprehensive prevention and treatment strategies for this disease.

The presence of excess protein in the urine, known as proteinuria, is recognized as a risk factor for cardiovascular diseases and mortality [6,7,8]. Proteinuria has been found to be associated with both stroke and coronary heart disease events, independent of other cardiovascular risk factors [9,10]. In addition, research indicates that proteinuria is a predictor of future risk for various diseases, such as hypertension, diabetes mellitus, and heart failure [11]. In particular, proteinuria has been identified as a significant determinant of the development of hypertension in Asian populations [12]. However, proteinuria can develop over time or may be reversible with risk factor modification or treatment. This suggests that proteinuria may be a modifiable factor in the development of hypertension. Despite this, to date, no large-scale study has evaluated whether the risk of developing hypertension changes in response to improvements in or the persistence of proteinuria.

Our hypothesis is that the risk of developing hypertension may differ depending on the persistence or resolution of proteinuria. To investigate this, our study aimed to explore the association between changes in proteinuria status and the subsequent risk of developing hypertension in a large-scale, nationwide longitudinal study of a general population. Through this investigation, our objective was to ascertain whether managing proteinuria status could potentially serve as an indicative approach for managing the risk of developing hypertension.

## 2. Materials and Methods

### 2.1. Database

The Korean National Health Insurance System (NHIS) provides comprehensive coverage of demographic and socioeconomic data, as well as a medical database for diagnosis and treatment modalities. Moreover, the NHIS provides a comprehensive health examination database and a medical care institution database [13]. NIHS subscribers are encouraged to undergo standardized medical health examinations every two years [14]. For this study, we enrolled participants from the National Health Insurance System-National Health Screening Cohort (NHIS-HEALS) [15] which comprised individuals who participated in medical health screening programs. Demographic details, including height, weight, household income, smoking status, alcohol consumption, exercise habits, and comorbidities, were collected from these individuals. The Institutional Review Board of Ewha Womans University College of Medicine (2023-5-20) approved the analysis and provided a consent waiver. The study was conducted according to the guidelines of the Declaration of Helsinki.

### 2.2. Study Population and Variables

We analyzed a total of 1,878,329 participants who underwent two consecutive health examinations during the first period (2003–2004) and second period (2005–2006). In order to ensure the integrity and reliability of the data, participants with missing information for the variables of interest were excluded from the analysis, resulting in the removal of a total of 212,838 individuals. Additionally, individuals with a documented history of hypertension prior to the study were also excluded from the analysis, resulting in the exclusion of 729,768 participants. After these exclusions, a final sample size of 935,723 participants were included (Figure 1).

As part of the comprehensive health screening program, morning urine samples obtained after overnight fasting were subjected to dipstick urinalysis to detect proteinuria. The dipstick results were assessed using a color scale that categorized proteinuria as ‘negative’, ‘1+’, ‘2+’, ‘3+’, or ‘4+’. For this study, we categorized the dipstick proteinuria results into two groups: ‘no proteinuria (-)’ and ‘overt proteinuria (≥1+)’. Following the classification of proteinuria status, the participants were further divided into four specific groups based on the presence or absence of proteinuria in the two consecutive health examinations: the (1) proteinuria-free (participants who exhibited no evidence of proteinuria during either the first or second screening), (2) proteinuria-resolved (participants who displayed evidence of proteinuria during the initial screening but subsequently demonstrated its resolution by the time of the second screening), (3) proteinuria-developed (participants who initially exhibited no proteinuria during the first screening but presented with positive results during the second screening), and (4) chronic proteinuria (participants who had evidence of proteinuria throughout both screening periods) groups.

We defined the index date as the date of the latest health examination and used the baseline characteristics from this examination. The outcome was the occurrence of hypertension. According to guidelines for management of hypertension, the definition of optimal blood pressure (BP) was a systolic BP (SBP) <120 mmHg and/or a diastolic BP (DBP) <80 mmHg [16,17]. New-onset hypertension was designated as a primary or secondary diagnosis of hypertension (International Classification of Diseases (ICD)-10 codes I10–I11). Hypertension was defined according to prescription records for any antihypertensive agent for at least one claim per year added to a record of visiting an outpatient clinic or admission. In addition, at least one SBP measurement greater than 140 mmHg or DBP measurement greater than 90 mmHg from the NHIS-HEALS after the index date was considered new-onset hypertension. As the NIHS covers the entire population through a compulsory scheme, the study had no instances of follow-up loss.

To comprehensively assess the influence of various factors, several covariates were collected at the index date, ensuring a comprehensive analysis. These covariates encompassed a range of characteristics and variables including age, sex, body mass index, and household income. Moreover, participants provided information on smoking habits, alcohol consumption (reported as days per week), and regular physical exercise (measured by frequency per week) through self-administered questionnaires. Smoking status was categorized as none, former smoker, and current smoker. Comorbidities of individuals were identified via the following criteria. Diabetes mellitus was defined as satisfying one of the following criteria: (1) at least one claim of diagnostic codes (ICD-10 E11–14) prior to index date with the prescription of an antidiabetic agent, (2) two or more claims of diagnostic codes (ICD-10 E11–14) prior to index date, (3) fasting serum glucose level ≥ 7.0 mmol/L, or (4) self-reported diabetes mellitus in the questionnaire. Dyslipidemia was defined as satisfying one of the following criteria: (1) at least one claim of diagnostic codes (ICD-10 E78) prior to index date with the prescription of a dyslipidemia-related agent, (2) two or more claims of diagnostic codes (ICD-10 E78) prior to index date, or (3) total cholesterol ≥ 240 mg/dL. Cancer was defined as having one admission or at least three outpatient claims of diagnostic codes (ICD-10 C00–97) prior to index date with a specific registration code of ‘V027′ or ‘V193–4′. Renal disease was defined as two or more claims of diagnostic codes (ICD-10 N17-19, I12-13, E082, E102, E112, E132) prior to index date, or estimated glomerular filtration rate less than 60 mL/min/1.73 m^2^. The Charlson Comorbidity Index was defined according to a previous study [18,19,20,21,22].

### 2.3. Statistical Analysis

We compared the baseline characteristics of the four groups using the Chi-square test for categorical variables and analysis of variance for continuous variables. Bonferroni post hoc analysis was used to identify significant differences between individual groups. Continuous variables are presented as the mean ± standard deviation, and categorical variables are presented as numbers (percentages). To evaluate the relationship between changes in proteinuria status and the occurrence of hypertension, we used Kaplan–Meier survival curves with the log-rank test. To determine the hazard ratio (HR), Cox proportional hazard with adjustment for confounding variables regression was used. In the multivariable Cox regression analysis, age and sex were adjusted in model 1; age, sex, body mass index, household income, smoking habits, alcohol consumption, regular physical exercise, and comorbidities (diabetes mellitus, dyslipidemia, atrial fibrillation, cancer, and renal disease) were adjusted in model 2; and age, sex, body mass index, household income, smoking habits, alcohol consumption, physical activity, comorbidities (diabetes mellitus, dyslipidemia, atrial fibrillation, cancer, and renal disease), and Charlson Comorbidity Index were adjusted in model 3. The results of the Cox regression analysis are presented as HRs and 95% confidence intervals (CIs). The assumption of the proportionality of hazards was assessed using Shoenfeld’s residuals, and no significant deviations from this assumption were observed. For the subgroup analysis, a pairwise comparison analysis was performed to assess the altered hypertension risk for those who recovered from or developed proteinuria; proteinuria-resolved vs. proteinuria-free, proteinuria-developed vs. proteinuria-free, proteinuria-resolved vs. chronic proteinuria, and proteinuria-developed vs. chronic proteinuria. For sensitivity analysis, a multivariable analysis was conducted by excluding those who had developed hypertension within one year from the index date to reduce the risk of reverse causality (landmark analysis). Furthermore, we performed further sensitivity analysis after the exclusion of participants with increased risk of hypertension and proteinuria such as participants who were aged over 65, had a BMI over 25 (kg/m^2^), were current smokers, did not regularly exercise, or had renal diseases. All statistical analyses were performed using Statistical Analysis System software (SAS version 9.2, SAS Institute, Cary, NC, USA). All values with *p*-values < 0.05 were considered statistically significant.

## 3. Results

Of the total 935,723 participants, 919,044 (98.2%), 7488 (0.8%), 8205 (0.9%), and 986 (0.1%) were categorized into the proteinuria-free, proteinuria-resolved, proteinuria-developed, and chronic proteinuria groups, respectively. The median interval between the first and second health screening was 21.5 months (interquartile range, 11.1–25.5 months). The baseline characteristics according to changes in proteinuria status are shown in Table 1. The mean age of the overall participants was 40.96 ± 11.01 years, and 62.5% were male. The chronic proteinuria group had a higher proportion of male participants than the other groups, while the proteinuria-resolved group had a higher proportion of female participants compared with the other groups. The chronic proteinuria group had the highest mean body mass index and a higher proportion of current smokers than the other groups. In terms of comorbidities, the prevalence of diabetes mellitus, dyslipidemia, atrial fibrillation, cancer, renal disease, and a Charlson Comorbidity Index ≥2 was the lowest in the proteinuria-free group and the highest in the chronic proteinuria group (Table 1).

Over a median follow-up of 14.23 (interquartile range: 8.89–14.68) years, 346,686 (37.1%) cases of hypertension occurred. The Kaplan–Meier survival curves for the occurrence of hypertension according to changes in proteinuria status are shown in Figure 2. The risk of developing hypertension varied with changes in proteinuria status. The chronic proteinuria group showed the highest risk of developing hypertension throughout the follow-up period, followed by the proteinuria-developed group, proteinuria-resolved group, and proteinuria-free group (Figure 2). In the multivariate analysis (model 3), when compared with the proteinuria-free group, the risk of hypertension was higher in the proteinuria-resolved group (HR: 1.17, 95% CI: 1.13–1.21, *p* < 0.001), proteinuria-developed group (HR: 1.31, 95% CI: 1.26–1.35, *p* < 0.001), and chronic proteinuria group (HR: 2.10, 95% CI: 1.94–2.26, *p* < 0.001) (*p* for trend < 0.001, Table 2).

In a subsequent pairwise comparison, the proteinuria-resolved group showed a significantly lower risk of hypertension than the chronic proteinuria group (HR: 0.58, 95% CI: 0.53–0.63, *p* < 0.001). Moreover, the proteinuria-developed group showed a significantly higher risk of hypertension than the proteinuria-free group (HR: 1.04, 95% CI: 1.26–1.35, *p* < 0.001) in the multivariable analysis (model 3) (Table 3).

When analyzing the initial severity of proteinuria and the associated risk of hypertension, we found that any level of proteinuria, when compared with no proteinuria, was associated with an increased risk of developing hypertension (Appendix A). Furthermore, we found that the relationship between changes in proteinuria status and the risk of developing hypertension remained consistent regardless of the presence of renal disease (Table 4).

In the sensitivity analysis using the landmark method, an association between changes in proteinuria status and the risk of hypertension was consistently noted (Appendix A). In addition, the pairwise comparison results showed a consistent association, indicating that the proteinuria-resolved group had a relatively lower risk of hypertension than the chronic proteinuria group (HR: 0.64, 95% CI: 0.59–0.69, *p* < 0.001) (Appendix A). In a further sensitivity analysis, after excluding participants with an underlying risk of hypertension, the association of changes in proteinuria status with future hypertension was consistently noted (Appendix A).

## 4. Discussion

The findings of this study indicate that changes in proteinuria status are closely correlated with changes in the risk of developing hypertension. Specifically, the highest risk of hypertension was observed among individuals categorized as having chronic proteinuria. Furthermore, our study found that individuals who recovered from proteinuria exhibited a lower risk of developing hypertension compared with those with persistent proteinuria. In contrast, individuals with newly developed proteinuria had a higher risk of hypertension than those who were proteinuria-free. In addition, these associations were consistently observed across different renal disease statuses. This suggests that regardless of the presence or absence of underlying renal conditions, the relationship between proteinuria status and hypertension risk remains significant and influential.

Although previous studies have reported an association between proteinuria and cardiovascular outcome, whether changes in proteinuria are associated with the risk of hypertension remains unclear. A previous study reported that the urine protein/creatinine ratio was associated with SBP among patients with chronic kidney disease [23], while another cross-sectional study found that spot urine protein excretion was associated with both high SBP and DBP in patients with diabetes mellitus [24]. In a longitudinal study of 4428 participants without a history of heart disease, baseline proteinuria was found to be associated with an approximately twofold higher risk of developing hypertension [12]. However, these studies only measured proteinuria at baseline and did not consider changes in proteinuria status, which is readily affected by various factors, such as medication use and lifestyle modification [25,26]. Furthermore, despite some inconsistencies, a recent meta-analysis found that changes in proteinuria, rather than its presence at baseline, were associated with cardiovascular outcome [27]. Our study reinforced these findings and provides a more thorough understanding of the relationship between proteinuria and hypertension. In this population-based study, individuals with chronic proteinuria had approximately twice the risk of developing hypertension compared with those remaining proteinuria-free.

Furthermore, in addition to the previous findings, this study contributes valuable insights by highlighting a significant association between recovery from proteinuria and a reduced risk of developing hypertension. These findings suggest that individuals who effectively resolve proteinuria may potentially experience significant benefits for their cardiovascular health. Evidence from previous studies consistently emphasizes the role of various lifestyle factors in hypertension prevention. Maintaining a healthy diet, managing body weight, engaging in regular physical activity, and practicing moderate alcohol consumption have all been linked to a decreased risk of hypertension [5,28]. Furthermore, specific factors such as folate intake and the use of non-narcotic analgesia have been identified as additional modifiable risk factors for hypertension [29,30,31]. This study supports the notion that managing proteinuria status is an independent modifiable risk factor that can be modified to reduce the likelihood of developing hypertension. By establishing a clear association between changes in proteinuria and the risk of hypertension, this study highlights the importance of effective proteinuria management in hypertension prevention. These findings emphasize the significance of addressing proteinuria and implementing appropriate strategies for its management to promote better cardiovascular health and reduce the risk of hypertension.

The link between proteinuria and the risk of developing hypertension may be explained by several factors. Adiponectin, the hormone secreted by adipocytes, could be the link between proteinuria and hypertension. Adiponectin plays a crucial role in regulating glucose and fatty acid metabolism and insulin sensitivity and exerting anti-inflammatory and anti-atherogenic effects [32,33]. Notably, a previous study revealed a negative correlation between the plasma adiponectin concentration and urinary albumin secretion [34]. Interestingly, this study also demonstrated that treatment with adiponectin normalized albuminuria in adipocyte-knockout mice. Further investigation through electron microscopy revealed that low adiponectin levels are associated with podocyte dysfunction. As a result, decreased visceral adiponectin due to the accumulation of body fat could contribute to both kidney dysfunction and increased systemic oxidative stress, ultimately leading to hypertension [35]. Asymmetric dimethylarginine (ADMA) is another potential link between proteinuria and hypertension. ADMA is endogenously produced through metabolic processes from proteolysis of arginine-methylated proteins and present normally in human circulation; it has also been implicated in endothelial dysfunction through inhibiting nitric oxide [36,37]. Since the majority of methylarginines are eliminated through the urine, patients with kidney impairment may experience elevated levels of ADMA in circulation, which can contribute to endothelial dysfunction and subsequent hypertension [37]. Moreover, chronic inflammation potentially establishes the connection between proteinuria and hypertension. Prior research has demonstrated proteinuria’s role as a biomarker of chronic inflammation [38]. In addition, the involvement of monocytes and macrophages in the inflammatory cascade significantly contributes to hypertension development [39]. Thus, inflammation emerges as a pivotal factor shaping the relationship between proteinuria and hypertension. Accordingly, targeting the management of chronic inflammatory status to address proteinuria holds promise as a therapeutic approach to prevent hypertension.

This study has several limitations. First, it should be noted that important laboratory data, specifically on urine creatinine, glomerular filtration rate, and albumin concentrations, could not be collected in our study, as the Korean NHIS dataset began including this information in 2009. These data could have provided crucial insights into the precise dose–response relationship, enabling a more accurate assessment of the impact of various factors. Additionally, it is important to acknowledge that the dipstick used for proteinuria detection has known limitations when it comes to detecting low levels of albumin excretion. This limitation restricts the comprehensive evaluation of the influence of microalbuminuria on hypertension. Second, the medication history of the participants was not considered. The use of medications such as angiotensin-converting enzyme inhibitors and angiotensin receptor blockers could significantly affect the proteinuria status of the participants. The absence of this information undermines the overall assessment and interpretation of the results. Third, transient proteinuria, which can occur in up to 4% of young adults, was not considered [40]. It is noteworthy that circumstances such as acute infections and strenuous exercise can impact the occurrence of transient proteinuria. Consequently, patients with transient proteinuria should have been excluded from this study. Unfortunately, our study cohort is lacking in information concerning underlying infections and strenuous exercise. Despite the presumption that the majority of participants within this cohort are unaffected by this condition, the absence of this information has the potential to introduce bias or confounding variables into our findings [40]. Fourth, the study failed to record the underlying etiology of proteinuria, including various conditions such as glomerular disease, tumorous conditions, infections, and autoimmune diseases. Understanding the specific causes of proteinuria is crucial for a comprehensive analysis and for identifying potential associations with other health conditions. Fifth, the proteinuria status of the participants at the conclusion of the follow-up periods was unavailable, a factor that could have provided significant insights into clarifying the relationship between proteinuria and hypertension. Sixth, the cohort of participants consistently demonstrating proteinuria status was notably smaller than the proteinuria-free group, potentially affecting the study’s outcomes. Finally, the generalizability of the results may be limited due to the study’s focus on a specific population. The participants were exclusively drawn from an Asian population, and caution should be exercised when extrapolating these findings to other ethnic or geographic groups.

## 5. Conclusions

In conclusion, this study suggests that changes in proteinuria status may have a significant relationship with the subsequent risk of developing hypertension. Importantly, the data indicate that individuals who successfully recover from proteinuria demonstrate a notably reduced risk of developing hypertension in comparison to those with chronic proteinuria. These observations suggest that monitoring and managing proteinuria status may serve as a valuable surrogate marker for preventing the onset of hypertension. By recognizing the impact of proteinuria on cardiovascular health, healthcare professionals can implement proactive strategies to reduce hypertension risk and enhance overall well-being.

## Figures and Tables

**Figure 1 jpm-13-01414-f001:**
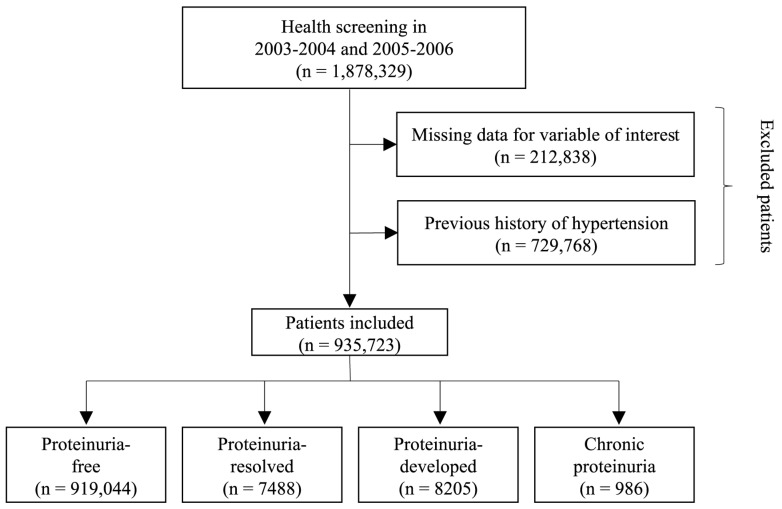
Flowchart of study participant selection.

**Figure 2 jpm-13-01414-f002:**
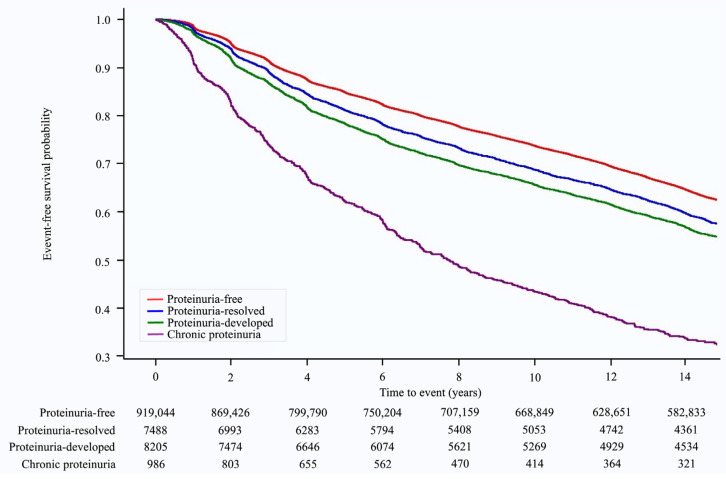
Kaplan–Meier survival curves for hypertension occurrence according to proteinuria status.

**Table 1 jpm-13-01414-t001:** Baseline characteristics of the study population.

	Total	Proteinuria-Free	Proteinuria-Resolved	Proteinuria-Developed	Chronic Proteinuria	*p*-Value
Number of participants (%)	935,723	919,044 (98.2)	7488 (0.8)	8205 (0.9)	986 (0.1)	
Age, years	40.96 ± 11.01	40.95 ± 10.99	41.67 ± 11.87	41.71 ± 11.77	43.43 ± 11.58	<0.001
Sex						<0.001
Men	585,143 (62.5)	576,080 (62.7)	3871 (51.7)	4493 (54.8)	699 (70.9)	
Women	350,580 (37.5)	342,964 (37.3)	3617 (48.3)	3712 (45.2)	287 (29.1)	
Systolic blood pressure (mmHg)	115.88 ± 8.82	115.88 ± 8.82	115.91 ± 9.25	115.95 ± 9.2	118.13 ± 9.09	<0.001
Diastolic blood pressure (mmHg)	72.85 ± 6.14	72.85 ± 6.13	72.65 ± 6.32	72.85 ± 6.36	74.26 ± 6.26	<0.001
Body mass index (kg/m^2^)	22.91 ± 2.86	22.91 ± 2.85	22.95 ± 3.01	22.94 ± 3.13	23.66 ± 3.27	<0.001
Household income						<0.001
Q1, lowest	133,759 (14.3)	131,192 (14.3)	1219 (16.3)	1219 (14.9)	129 (13.1)	
Q2	364,149 (38.9)	357,587 (38.9)	2962 (39.6)	3281 (40.0)	319 (32.4)	
Q3	304,680 (32.6)	299,626 (32.6)	2247 (30.0)	2465 (30.0)	342 (34.7)	
Q4, highest	133,135 (14.2)	130,639 (14.2)	1060 (14.2)	1240 (15.1)	196 (19.9)	
Smoking						<0.001
Never	564,164 (60.3)	553,337 (60.2)	4914 (65.6)	5388 (65.7)	525 (53.3)	
Former	104,020 (11.1)	102,352 (11.1)	757 (10.1)	772 (9.4)	139 (14.1)	
Current	267,539 (28.6)	263,355 (28.7)	1817 (24.3)	2045 (24.9)	322 (32.7)	
Alcohol consumption (days/week)						<0.001
<3	661,033 (70.6)	648,951 (70.6)	5464 (73.0)	5963 (72.7)	655 (66.4)	
≥3	274,690 (29.4)	270,093 (29.4)	2024 (27.0)	2242 (27.3)	331 (33.6)	
Regular exercise (days/week)						<0.001
<3	772,628 (82.6)	759,047 (82.6)	6067 (81.0)	6720 (81.9)	794 (80.5)	
≥3	163,095 (17.4)	159,997 (17.4)	1421 (19.0)	1485 (18.1)	192 (19.5)	
Comorbidities						
Diabetes mellitus	83,324 (8.9)	81,058 (8.8)	967 (12.9)	1089 (13.3)	210 (21.3)	<0.001
Dyslipidemia	166,485 (17.8)	162,667 (17.7)	1658 (22.1)	1837 (22.4)	323 (32.8)	<0.001
Atrial fibrillation	963 (0.1)	940 (0.1)	11 (0.2)	10 (0.1)	2 (0.2)	<0.001
Cancer	14,261 (1.5)	13,930 (1.5)	162 (2.2)	149 (1.8)	20 (2.0)	<0.001
Renal disease	2685 (0.3)	2532 (0.3)	76 (1.0)	48 (0.6)	29 (2.9)	<0.001
Charlson Comorbidity Index						<0.001
0	400,108 (42.8)	393,485 (42.8)	2961 (39.5)	3302 (40.2)	360 (36.5)	
1	395,345 (42.3)	388,548 (42.3)	3104 (41.5)	3339 (40.7)	354 (35.9)	
≥2	140,270 (15.0)	137,011 (14.9)	1423 (19.0)	1564 (19.1)	272 (27.6)	

Data are presented as the mean ± standard deviation or number (percentage). BMI, body mass index; Q, quartile.

**Table 2 jpm-13-01414-t002:** Multivariable Cox analysis for incident hypertension according to changes in proteinuria status.

Group	Total (n)	Hypertension (n)	Incidence Rate (per 1000 Person Years)	HR (95% Confidence Interval)		
				Model 1	Model 2	Model 3
Proteinuria-free	919,044	339,260	31.5	1 (ref)	1 (ref)	1 (ref)
Proteinuria-resolved	7488	3131	37.4	1.19 (1.15, 1.23)	1.17 (1.13, 1.21)	1.17 (1.13, 1.21)
Proteinuria-developed	8205	3638	41.3	1.31 (1.27, 1.35)	1.31 (1.27, 1.35)	1.31 (1.26, 1.35)
Chronic proteinuria	986	657	81.4	2.61 (2.41, 2.81)	2.11 (1.95, 2.27)	2.09 (1.94, 2.26)
*p* for trend				<0.001	<0.001	<0.001

Model 1 was adjusted for age and sex. Model 2 was adjusted for age, sex, body mass index, household income, smoking, alcohol consumption, physical activity, history of diabetes mellitus, dyslipidemia, atrial fibrillation, cancer, and renal disease. Model 3 was adjusted for age, sex, body mass index, household income, smoking, alcohol consumption, physical activity, history of diabetes mellitus, dyslipidemia, atrial fibrillation, cancer, renal disease, and Charlson Comorbidity Index. HR, hazard ratio; CI, confidence interval.

**Table 3 jpm-13-01414-t003:** Pairwise comparisons of the association between the change in proteinuria status and the risk of incident hypertension.

	Model 1		Model 2		Model 3	
	HR (95% CI)	*p*-Value	HR (95% CI)	*p*-Value	HR (95% CI)	*p*-Value
Proteinuria-resolved vs. Proteinuria-free (reference)	1.19 (1.15, 1.23)	<0.001	1.17 (1.13, 1.21)	<0.001	1.17 (1.13, 1.21)	<0.001
Proteinuria-developed vs. Proteinuria-free (reference)	1.31 (1.27, 1.36)	<0.001	1.31 (1.27, 1.35)	<0.001	1.31 (1.26, 1.35)	<0.001
Proteinuria-resolved vs. Chronic proteinuria (reference)	0.46 (0.43, 0.50)	<0.001	0.58 (0.53, 0.63)	<0.001	0.58 (0.53, 0.63)	<0.001
Proteinuria-developed vs. Chronic proteinuria (reference)	0.52 (0.48, 0.56)	<0.001	0.65 (0.59, 0.70)	<0.001	0.65 (0.60, 0.71)	<0.001

Model 1 was adjusted for age and sex. Model 2 was adjusted for age, sex, body mass index, household income, smoking, alcohol consumption, physical activity, history of diabetes mellitus, dyslipidemia, atrial fibrillation, cancer, and renal disease. Model 3 was adjusted for age, sex, body mass index, household income, smoking, alcohol consumption, physical activity, history of diabetes mellitus, dyslipidemia, atrial fibrillation, cancer, renal disease, and Charlson Comorbidity Index. HR, hazard ratio; CI, confidence interval.

**Table 4 jpm-13-01414-t004:** Subgroup analysis of multivariable Cox analysis for incident hypertension according to changes in proteinuria status.

**With Renal Diseases**				
**Proteinuria Status**	**Total (n)**	**Hypertension (n)**	**Incidence Rate (per 1000 Person Years)**	**Adjusted HR (95% CI) ***
Proteinuria-free	2532	1562	70.94	1 (ref)
Proteinuria-resolved	76	56	98.21	1.46 (1.12, 1.91)
Proteinuria-developed	48	33	88.31	1.43 (1.01, 2.02)
Chronic proteinuria	29	25	185.34	2.62 (1.76, 3.91)
*p* for trend				<0.001
**Without Renal Diseases**				
**Proteinuria Status**	**Total (n)**	**Hypertension (n)**	**Incidence Rate (per 1000 Person Years)**	**Adjusted HR (95% CI) ***
Proteinuria-free	916,512	337,698	31.42	1 (ref)
Proteinuria-resolved	7412	3075	37.02	1.16 (1.12, 1.21)
Proteinuria-developed	8157	3605	41.05	1.30 (1.26, 1.35)
Chronic proteinuria	957	632	79.61	2.08 (1.92, 2.25)
*p* for trend				<0.001

* Adjusted for age, sex, body mass index, household income, smoking, alcohol consumption, physical activity, history of diabetes mellitus, dyslipidemia, atrial fibrillation, cancer, renal disease, and Charlson Comorbidity Index. HR, hazard ratio; CI, confidence interval.

## Data Availability

The data that support the findings of this study are available from NHIS-HEALS, but restrictions apply to the availability of these data, which were used under license for the current study and are thus not publicly available. However, these data are available from the authors upon reasonable request and with permission from the National Health Insurance System.

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
