# Peer review of "Association between Proteinuria Status and Risk of Hypertension: A Nationwide Population-Based Cohort Study"

_jpm, 2023, doi:10.3390/jpm13091414_

Round 1
Reviewer 1 Report
In the manuscript titled “Association between proteinuria status and risk of hypertension: A nationwide population-based cohort study”, Dr. Hyungwoo Lee and team sought to understand whether changes in proteinuria status between a first and second exam impact the incidence of hypertension using the Korean National Health Database. The paper is well-written and shares some interesting insights, however I share my concerns below.
Major concerns:
1. Some comorbidities are associated with increased risk of hypertension and proteinuria, such as overweight/obesity status, sedentary lifestyle, smoking status, age over 65 and history of renal pathologies-- why were these not excluded from the study? Table 1 in fact shows that the chronic proteinuria has higher prevalence of these, which would confound the results even if these were adjusted for, and thus is not a fair assessment of the incidence of hypertension. Were there any sensitivity analyses with the exclusion of these cases performed?
2. There is a 900:1 difference between proteinuria-free and chronic proteinuria groups (Figure 1), making the latter group extremely underpowered.
3. The clinical characteristics table (Table 1) needs more information, such as systolic and diastolic blood pressure as well as glomerular filtration rate (GFR), if available. All data should also be provided for the 14.2-year follow-up.
4. Was the proteinuria status obtained after the 14.2-year follow-up? I think evaluating the incidence/update of proteinuria status, its associated with the initial changes in proteinuria in exam 1 vs exam 2, and the development of hypertension would be meaningful to share.
5. Not being able to explain transient proteinuria or the underlying causes of the proteinuria is something of concern. Can you please elaborate on why the protein-resolved and protein-developed cases would be included here?
Minor concerns:
1. Please include mean age, sex and hypertension incidence among the participants in the abstract
2. Please state the purpose of the study (aim/objective) of the study in abstract.
3. Table 1 columns seem to be improperly labeled (three columns labeled as “Proteinuria-“)- please address this.
4. The 14.2-year follow-up, along with the loss to follow-up information is missing from the methods-- please address this.
5. Were there any cases of acute reactionary proteinuria among the patients? You noted that transient proteinuria was not considered, but an underlying infection would be an example of this and thus should be excluded. In essence, were the participants relatively healthy upon assessments?
Reviewer 2 Report
Main issue with the paper titled "Association between proteinuria status and risk of hypertension: A nationwide population-based cohort study" is detection of proteinuria with dipsitick method rather than determining protein excretion in 24 h urine sample. Confounding factors such as infection, strenuous exercise should be excluded. There are no other methodological concerns.
Discussion should focus on inflammation which is a part of both proteinuria and hypertension. Moreover, clinical usage of the study findings can be emphasized in this section. Otherwise discussion is fine.
